



**Total column ozone in New Zealand and in the UK in the 1950s**
**Stefan Brönnimann[1,2] and Sylvia Nichol[3]**
[1] Oeschger Centre for Climate Change Research, University of Bern, Switzerland
[2] Institute of Geography, University of Bern, Switzerland
[3] National Institute for Water and Atmospheric Research (NIWA), Wellington, New
Zealand
**Abstract**
Total column ozone measurements reach back almost a century. Historical column ozone data
are important to obtain a long term perspective of changes of the ozone layer, but arguably
also as diagnostics of lower stratospheric or tropopause-level flow in time periods of sparse
upper-air observations. With the exception of few high quality records such as that from
Arosa, Switzerland, ozone science has almost exclusively focused on data since the
International Geophysical Year (IGY) in 1957, although earlier series exist. In the early
2000s, we have digitised and re-evaluated many pre-IGY series. Here we add a series from
Wellington, New Zealand, 1951-1959. We re-evaluated the data from the original observation
sheets, performed quality control analysis and present the data. The day-to-day variability can
be used to assess the quality of reanalysis products, since the data cover a region and time
period with only few upper-air data. Comparison with total column ozone in the reanalyses
ERA-PreSAT (which assimilates upper-air data), 20CRv3 and CERA20C (which do not
assimilate upper-air data) shows high correlations with all three. Although trend quality is
doubtful (no calibration information and no intercomparisons are available), combining the
record with other available data (including historical data from Australian locations) allows a
70-year perspective of ozone changes over the southern midlatitudes. The series is available
from the World Ozone and Ultraviolet Data Centre. Finally, we also present a short series
from Downham Market, UK, covering November 1950 to October 1951, and publish it with
further historical data series that were previously described but not published.

**1 Introduction**
Regular total column ozone measurements reach back almost a century (Fabry and Buisson,
1921; Dobson and Harrison, 1926). While interest first arose from its close relation to





tropopause flow, which seemed promising as a meteorological diagnostic prior to the
invention of the radiosonde, the focus then shifted towards understanding stratospheric
circulation and monitoring of the ozone layer. Historical data were not considered particularly
important until the onset of ozone depletion and the discovery of the Antarctic ozone hole.
Even then, the focus was on ozone changes since the International Geophysical Year (IGY) in
1957/58, when a global network was initiated and a new measurement protocol (double
wavelength pair) was introduced, leading to higher quality measurements (Dobson, 1957a,b;
Dobson and Normand, 1957). Only few, long records such as the one from Arosa were re-
evaluated (Staehelin et al., 1998), providing an important basis for trend assessments (see also
Müller, 2009 and Bojkov, 2012, for a history of ozone measurements).
In the early 2000s, the first author compiled and digitised a considerable number of pre-IGY
series in order to exploit their relation to tropopause flow and the stratospheric meridional
circulation (Brönnimann et al., 2003a,b). Trend quality is not necessarily required for such
applications since the day-to-day variation at mid-latitudes is much larger than the trend. The
data were digitised, homogenised if possible and some (but not all) were delivered to the
World Ozone and Ultraviolet Data Centre (WOUDC). Not all existing series could however
be found. Here we add further series to this collection, namely from Wellington, New
Zealand, 1951-1959 (the data from the IGY onward are already in the WOUDC data base)
and a short and patchy series from Downham Market, UK, from November 1950 to October
1951. In this paper we present the series, their quality control and show selected analyses. The
data are used to independently assess reanalysis data sets, and the long term changes of ozone
over the southern midlatitudes since the 1950s is presented.
The paper is organised as follows. Section 2 presents the instrument history and Section 3
describes the data re-evaluation. Comparisons with upper-air data and reanalysis data sets are
presented in Section 4. In Section 5 we provide an assessment of the data quality and compare
the results with literature. Conclusions are drawn in Section 6.

**2. Ozone data and instrument histories**
*2.1. Wellington (D#17)*
Already during Dobson's first (photographic) global ozone network in the late 1920s (Dobson
et al., 1930), New Zealand participated by hosting a spectrophotometer in Christchurch (Fig.
1). When Dobson built the new photoelectric instruments in the 1930s (Dobson, 1931) and
planned a global network with these instruments, New Zealand was approached again and in



1937 eventually placed an order (see Nichol, 2018; Farkas, 1954). However, delays occurred,
and the designated instrument (Dobson #17) was only finished shortly before the war. When
the war started, the UK approached New Zealand and asked to withhold the delivery of D#17
in order to use it in the UK. The instrument operated in the UK until 1947. It was then
decided that a recalibration and improvement was necessary before the instrument could be
shipped to New Zealand, therefore, the instrument was sent to Oxford. The photoelectric cell
and amplifier were replaced by a photomultiplier (Farkas, 1954). In Dobson's original
observation sheets from Oxford (Vogler et al., 2007) we found measurements performed with
D#17 on 24 Feb and 1 Mar 1940 and then again on 21 and 22 Nov 1946. This was
presumably before the upgrade. Note, however, that these observation sheets are incomplete.
No sheets from Oxford could be found for the period from January 1947 to October 1949,
which might have contained the calibration information (together with other measurements
from Oxford, which are lost).
The instrument was sent from the UK only in late 1949 and arrived in New Zealand in 1950.
The instrument was first tested, and it was found that the setting of the quartz plates had
altered during the transport (Farkas, 1954). As a consequence, a new table of plate settings
was produced for operations. Then the instrument was put in operation in Kelburn,
Wellington (Fig. 1).
The first measurements are dated 1 August 1951. In the first years, Elizabeth Porter was in
charge of the measurements. After her unexpected death in 1953, Edith Farkas took over and
was in charge of operations until the mid-1980s. The instrument underwent another major
rehaul in 1963/4. At this occasion it was also compared with D#105 (Nichol, 2018).
For all observations, the shorter wavelength was 311.2 nm (C pair) and measurements were
taken in direct sun (DS) mode as well as at the blue (ZB) or cloudy zenith (ZC, using an
additional wavelength that is not strongly absorbed by ozone; the pair formed by the two
longer wavelengths, sometimes termed C', allows addressing the attenuation by clouds). The
relative path length through the ozone layer, μ, was calculated from a nomogram. The altitude
of the ozone layer was assumed to be 22 km. For DS measurements, an atmospheric
correction was added, which was assumed to be 0.095 m atm. cm for clear days and 0.1 for
slightly hazy days and more (usually 0.11) for very hazy days.
Observations at the blue or cloudy zenith require calibration using quasi-simultaneous
observations. In 1954, when the report was published, only a limited set of such observations
was available, values were described as somewhat doubtful (Farkas, 1954). For this paper, we
thus recalibrated these measurements.





Farkas (1989) and Nichol (2018) consider the data prior to 1964 unreliable, as no
intercomparison had been made. For the sake of completeness, Nichol (2018) shows data
from the IGY onward, though noting their inferior quality. These data, from July 1957
onward, are available from the WOUDC. However, the data prior to 1957 have so far not
been available electronically. The earliest data have been published by Farkas (1954), where
in addition to the reduced ozone amount also the observation mode, wavelength pair used, and
observation time was indicated. Reduced values were sent to the International Ozone Office,
where the communication was stored and later sent to Environment Canada. It was scanned
and recently sent to the first author as a PDF file (Bais, personal communication).
We digitised the total column ozone data from both sources, the PDF file from the
International Ozone Office as well as from Farkas (1954). Upon inquiry, the original data
sheets (covering 1951 to 1960) were found at NIWA, scanned, and sent to the first author
(Fig. 2). The original readings were then also digitised. The main source of information in this
paper are the original sheets; the reduced values from the other two sources were used for
cross-checking. Note that we do not have calibration information or intercomparison data.
However, the data sheets contain many notes that provide additional information on the
instrument history. This information will be given in Sect. 3.

*2.3. Downham Market*
The scans from the Ozone Office also contained data from Downham Market, though almost
illegible. These are daily averaged, reduced total column measurements with no additional
information. They covered the year 1951 (January to October). We supplemented these data
with values printed on a graph (incidentally, this was a New Year's card sent out by the
International Ozone Office, Fig. 3), such that we could extend the series backward to late
November 1950. Note that both sources of information are secondary sources and thus
inherently unreliable. Nevertheless, as will be shown, the quality of the data seems
unexpectedly high.
Sometimes monthly means were indicated on the sheet, which we could use to cross check
our digitisation. Additionally, monthly data from Downham Market (November 1950 to
October 1951) were found in the communication of the International Ozone Office, stored at
the UK Met Office (Normand, 1961). These data were also used to cross-check where there
were no monthly means in the other source, although there were also sometimes differences
between the monthly means from both sources. This second source (Normand, 1961) also





showed us that the record would have continued into November 1951 for at least 17 days, and
that 15 and 26 daily values are missing in our source for November and December 1950,
respectively.
Nothing is known about the instrument or the history of the measurements. We assume that
the instrument (the number remains unknown) was relocated to Hemsby in November 1951.
Brönnimann et al. (2003b) digitised the Hemsby total column ozone data and found a good
quality (in terms of day-to-day changes) apart from an unplausible (flagged) period. The
context of the measurements remains also unknown. Scrase (1951) mentions the testing of
radiosondes at Downham Market in approximately the same period.

**3. Re-evaluation and analysis methods**
*3.1. General procedure*
The processing of Dobson data is described in Komhyr and Evans (2006); the standard
procedure to re-evaluate the data is given in Bojkov et al. (1993). We followed the two
guidelines as closely as possible. Note, however, that no calibration information and no
intercomparison data were available. The standard equation for calculating total column
ozone $X$ (in atm. cm at standard pressure) from a single wavelength pair (with short and long
wavelengths $\lambda$ and $\lambda'$) is:
$$X = \frac{N - (\beta - \beta')\dfrac{mp}{p_0} - (\delta - \delta')\sec(SZA)}{(\alpha - \alpha')\mu}$$    (Eq. 1)
where $\beta$ is the molecular scattering coefficient (primes denote the longer wavelength), $\alpha$ is the
absorption coefficient, $\delta$ is the aerosols scattering coefficient, $m$ is the relative air mass, $\mu$ is
the relative path length through the ozone layer, and $SZA$ is the solar zenith angle. The relative
intensity $N$ is the actual measurement:
$$N = \log\left(\frac{I_0}{I_0'}\right) - \log\left(\frac{I}{I'}\right)$$    (Eq. 2)
where $I$ and $I_0$ are the intensities at the surface and outside the Earth's atmosphere,
respectively. $N$ is obtained from the dial reading at the instrument, $R$, via a conversion table
($R$-$N$ table). No unique value can be given for the aerosol scattering coefficient ($\delta$-$\delta'$) as it
depends on the haziness of the atmosphere.
For double wavelength pairs such as AD or BD, the following equation is used:





$$X_{12} = \frac{(N_1 - N_2) - [(\beta - \beta')_1 - (\beta - \beta')_2]\frac{mp}{p_0} - [(\delta - \delta')_1 - (\delta - \delta')_2]\sec(SZA)}{[(\alpha - \alpha')_1 - (\alpha - \alpha')_2]\mu}$$ (Eq. 3)
Aerosol scattering can then be neglected.
When re-evaluating historical data, the procedure is to first process the DS data (the double
pair data can be processed directly, while the single pair data require assumptions concerning
aerosol scattering). The ZB observations are then calibrated against quasi-simultaneous
(typically within minutes) DS observations by fitting $N$ and $\mu$ using third order polynomials
(Vanicek et al., 2003):
$X = c_0 + c_1 N + c_2 \mu + c_3 N^2 + c_4 \mu^2 + c_5 N^3 + c_6 \mu^3 + c_7 N \mu + c_8 N \mu^2 + c_9 N^2 \mu$   (Eq. 4)
Vanicek et al. (2003) recommend to split the data into seasons and fit polynomial functions
separately.
In a second step, ZC observations are processed. This is done by adjusting $N$ by adding a term
$\Delta N$ in such a way that they can be processed similar to ZB observations. For the C pair, $\Delta N$ is
determined by means of an additional wavelength pair, C', the shorter wavelength of which
corresponds to the longer wavelength of the C pair. Both wavelengths of the C' pair are very
little absorbed by ozone and thus allow assessing the aerosol and cloud scattering. The
correction additionally depends on the cloud type and altitude. Vanicek et al. (2003) use cloud
attenuation tables for the correction; constructing such a table however requires a lot of
parallel measurements. Vogler et al. (2006) uses linear regressions of the form
$\Delta N = c_0 + c_1 N_{C'}$                                                              (Eq. 5)
separately for situations with high clouds and situations with middle or low clouds. Here, $\Delta N$
is the difference between $N$ of a quasi-simultaneous ZB measurement and $N$ of the ZC
measurement (both for the C pair), while $N_{C'}$ refers to the C' pair of the ZC measurement.
If original observations sheets are not available, all that can be used are the calculated total
column ozone values as well as, if available, the time of day (which allows calculating SZA).
Changes in the absorption scale can be corrected by scaling the data (see Brönnimann et al.,
2003b) and statistical corrections must be used otherwise. Assessing the dependence of, e.g.,
differences to a neighbouring station, on SZA or on the annual cycle can give some hints on
possible causes for biases. Statistical corrections can be made dependent on the seasonal cycle
or SZA, although series processed in this way are likely to be of a lower quality.
In this paper we followed the former, detailed approach for Wellington and the latter approach
for Downham Market. The following sections describe the details of the processing.





*3.2. Wellington*
All observations, 2500 in total, were digitised. Zenith observations were noted on the sheet
but the distinction between ZB and ZC is not made on the sheets until 1954 (however, prior to
that time the observations and calculations indicate whether a zenith observations was
performed at the clear or cloudy zenith, and some of the measurements could be double
checked with Farkas, 1954). ZC observations were performed from the beginning, often in
pairs (ZB and DS, ZC and DS). Observation pairs of ZB/ZC or observation triplets only
follow later. From 1955 onward, there are occasional observations of the A pair, and from
1957 on of the AD pair. In 1957 numerous quasi-simultaneous observations of AD and C
pairs were performed, then AD measurements were no longer performed, while BD
measurements became frequent.
There are almost no measurements from July 1956 to February 1957, which is also confirmed
in the data from the Ozone Office. The second half of 1958 was missing entirely from the data
sheets, but in that case daily data were sent to the Ozone Office and are today found at
WOUDC, indicating that data sheets have been lost. Our material continues in January 1959.
From September 1959 onward, various problems seemed to have occurred, according to notes
on the observation sheets. One note reads: "While putting lid back after battery change on 8
October 1959, the quartz plates must have moved. From standard lamp readings the estimated
correction for dial readings is as follows: b + 9, c+c' + 6, d + 10". Another note in October
1959 speculated that "Quartz plates might have moved at beginning of September at one of
the occasions when silica gel was changed". From October 1959 onward, data sheets become
relatively messy, with black ink, red pencil, and many strike throughs. It is hard to follow if
and which corrections were done. A deterioration was also found in terms of correlation and
was visually apparent when plotting the data. Problem with the quartz plates are also
mentioned later on (e.g., an adjustment in February 1960 is mentioned). We therefore only
consider data prior to September 1959.
From the original observations we basically used only the dial readings $R$ and the time of
observations as well as information on the haziness and cloud cover, but all other calculations
were nevertheless digitised and provided important information. For instance, we checked the
averaging of the different $R$ readings, we reassessed the $R$-$N$ conversion (which is a linear
function per wavelength) and found that the relation has not changed over the period under
study. In this way we checked all steps of the original calculations, where possible.
Inconsistencies led to the correction of digitisation errors, of typos on the original sheets, or of
miscalculations; however, some could not be resolved and led to the flagging of observations.



From the time we calculated the solar zenith angle SZA using the MICA software. The
variables $m$ and $\mu$ (assuming an ozone layer height $h$ of 22 km) were calculated from SZA
following Komhyr and Evans (2006). We extracted sea-level pressure from the Twentieth
Century Reanalysis version 3 (20CRv3, Slivinski et al., 2019) and calculated station pressure
$p$ assuming a gradient of 0.125 hPa m$^{-1}$. Note that we could also have used the original $\mu$
calculations and neglected the pressure dependence. The effect of each of these factors is ca.
1-2 DU (referring to the standard deviation; this is much smaller than the observation error).
Our procedure allowed further checks and thus further corrections of erroneous data, though it
might also have introduced further errors (e.g., digitisation errors of the time of day).
According to Farkas, the shorter wavelength of the C pair was 311.2 nm, which slightly
deviates from the nominal value of 311.45 nm for the C wavelength pair. Therefore, we tested
two sets of absorption coefficients: the standard Bass-Paur absorption coefficients (Komhyr et
al., 1993) as well as modified coefficients. Using the standard coefficients can be justified by
the fact that we do not know the slit function for this specific instrument. Furthermore, the full
width-at-half-maximum is typically larger than 1 nm, such that effects are likely small.
Modified coefficients can be motivated by the work of Svendby (2003), who adjusted
coefficients for D#8 with a centre wavelength of 311.0 nm (she could actually measure the slit
function of D#8). As an approximation, we can interpolate between her value and the Bass-
Paur coefficient, yielding $\alpha = 0.891$. Assuming that the long wavelength was the same, we get
$(\alpha-\alpha')$ of 0.851. Similarly, the Rayleigh scattering coefficient was adjusted and $(\beta-\beta')$ was set
to 0.111.
In the calculation sheet sent to observers in the 1950s, molecular and aerosol scattering were
not distinguished. Only the first term of the equation, $N / (\alpha - \alpha') \mu$, was evaluated. From this,
Dobson suggested to subtract 95 DU on clear days and 100 DU (occasionally more) on hazy
days. Using Eq. 1 we can calculate molecular scattering and find that it amounts to ca. 95 DU,
leaving 0 to 15 DU to aerosols, depending on haziness. Svendby (2003), for a site in Norway,
found aerosol scattering contributions of 0 to 4% using direct sun C' observations. In order to
determine aerosol scattering we analysed all CC' observations performed in DS mode. Only
23 observations were found, and using the method of Svendby (2003) we found inconsistent
results (negative coefficients), indicating that the longer wavelength of the C' pair might have
been different from that in D#8. We therefore assumed an aerosol scattering coefficient $(\delta-\delta')$
for the C pair of 0.001 for clear days (the vast majority of days), 0.005 for hazy days and 0.01
for very hazy days. This is less than indicated in the tables that came with the instrument
D#42 in College, Alaska, for which we have the numbers (0.006, 0.018, 0.029 for slightly





hazy, hazy, and very hazy days, respectively; see Brönnimann et al. 2003b). However, the
coastal station Wellington might be less affected by aerosols than Oxford or College. Our
correction corresponds to aerosol effects of ca. 1.2, 6, and 12 DU which is consistent with
Svendby (2003) and also yields consistent results between C and double-wavelength pair
measurements (see below).
We then processed all DS data. AD DS measurements have become the standard with the
IGY. However, the correlation of AD DS total ozone with the C DS data was very low
(around 0.5) and the seasonal cycle of AD DS measurements was unrealistic. Obviously there
was a problem with the A wavelength pair, and this must have been the reason why AD
measurements were discontinued and BD measurements were performed later on. Therefore,
we did not further pursue A and AD measurements.
We then compared the BD DS data with quasi simultaneous (<3 hr time difference) C DS data
(Fig. 4a). We identified 136 pairs, and their correlation was 0.85. The C DS measurements are
slightly lower than the BD DS measurements (by 1.8%) when adjusted coefficients are used,
slightly higher (1.0%) when Bass-Paur coefficients are used.
In the next step we compared the C DS data with quasi simultaneous (<3 hrs) C ZB data. We
identified 429 pairs and applied Eq. 4, stratifying the data into May to October and November
to April, respectively. We found an overall good fit (Fig. 4b), with explained variances of
87% and 95% for the two seasons, respectively (numbers are the same for Bass-Paur or
adjusted coefficients). The standard deviations of the residuals were 12 DU for the winter and
9 DU for the summer season.
Next we compared C ZB with C ZC data. We found only 65 quasi-simultaneous observations
(Fig. 4c). Separating them into different cloud types was impossible as almost all
measurements were for cumulus. We therefore fit only one function, but rather than a linear
function as in Vogler et al. (2006) we used a second order polynomial function. The explained
variance of the fit $R^2$ was 0.58. The corrections for $N$ that were obtained in this step were then
applied to the Z ZC data and they were then reduced with the same equation as the C ZB data.
As a further test we then selected quasi-simultaneous (<3 hrs) observations of C DS and C ZC
and found 178 pairs (Fig. 3d). The correlation was 0.96 and the standard deviation of the
differences amounted to 13 DU, but a mean bias of 5.8 DU (5.7 DU for the case with adjusted
coefficients) is apparent. We therefore subtracted 5.8 DU (5.7 DU) from all ZC observations.
In this way all data could be processed. During the process we discovered sometimes
inconsistences (e.g., errors in the calculation performed in the 1950s, or typos), and some



values were marked with question marks on the sheets. While some of the problems (e.g.,
miscalculations or typos) could be resolved, in other cases such values were flagged in our
data set, though we still reduced the ozone amount. We also flagged other suspect values, e.g.,
cases where $N$ values were not reduced at all on the sheets. In total, of the 2500 observations
digitised, 2253 values were reduced, of which 56 were flagged. By definition of the
procedure, DS data are the reference, while ZB data and ZC data are fitted to the DS data in
two steps and thus a somewhat lower quality is expected.
Finally, we compared our reduced values to those digitised from the Ozone Office files as
well as to those stored at WOUDC. This revealed further important information. For instance,
January and February 1959 are missing in the Ozone Office data but not in our data sheets.
The non-reporting could be due to low quality. In fact, many values in January 1959 had
question marks on the original sheets and there is a note that the battery was extremely low;
on 4 February battery and spring were replaced and the rhodium plate was fixed to position
"opaque". In our series, however, only a sequence of values in January 1959 was flagged.
For further comparisons we averaged our values (not considering flagged values) to daily
means using New Zealand dates as well as UTC dates and then compared with the two daily
data sets. Both sources (Ozone Office, WOUDC) used New Zealand dates, although both are
shifted by one day after February 1959. We found a generally good agreement; discrepancies
were checked, which led to the flagging of two additional values, while most checked values
were not flagged.
Finally, for the daily data set, we supplemented the missing half year in 1958 with the data
from the Ozone Office, scaled with 1.041 to account for the change in absorption coefficients.
All processed original observations as well as the supplemented daily values are shown in
Figure 5 (here we show the version with Bass-Paur coefficients). No obvious discrepancies
are found, although the scatter in the C ZC data is visibly larger than for C DS or C ZB data.
In this way the data set is used in the following.

*3.3. Downham Market*
In the case of Downham Market, our data are only daily mean, reduced total column
measurements. All that can be done is to adjust them to account for the change in the
absorption cross sections used. At the time of the measurement, the so-called Ny-Choong
scale was in use. With the IGY, the Vigroux (1953) scale was adopted, but a few years later
was found to provide inconsistent results and was replaced by an updated Vigroux scale.





Finally, the Bass-Paur scale was adopted as standard (Komhyr et al., 1993). To convert
directly from the Ny-Choong to the Bass-Paur scale, we multiplied the all values with 1.416,
as recommended in Brönnimann et al. (2003b).
Several daily values were illegible, and two were marked with a question mark on the sheet
and were correspondingly flagged. The monthly mean values were used to cross-check the
numbers. The digitised raw data were then compared with the data from Oxford (Vogler et al.,
2007). Using linear regression with Oxford total column ozone as an independent variable,
days with exceedingly large residuals (outside ±3 standard deviations) could be flagged and
further checked (e.g., checking for digitising errors or by comparing the value with the days
before and after). Only one suspect measurement was found; it was flagged correspondingly.
A very high correlation of 0.91 was found between the series. Although the data only cover
one year, the difference series showed a clear seasonal cycle, with largest differences
approximately around summer solstice. Offsets that include a seasonal cycle are possible due
to effects that either depend on the solar zenith angle (e.g., due stray light in the instrument),
on temperature, on the ozone amount, or on the tropopause height. The data amount is not
sufficient to decide between different seasonalities. However, given the very high correlation
between the data from Downham Market and Oxford, pointing to a high day-to-day accuracy,
we adjusted the Downham Market data by subtracting a seasonal cycle based on fitting the
first harmonic to the difference series. Corrections are between 13 (winter) and 58 (summer)
DU.
Repeating the regression approach on this series we found one additional potential outlier
(outside ±3 standard deviations) that was correspondingly flagged. In this format the series is
used further in our paper.

*3.4. Comparison with other data sets*
In addition to Oxford total column ozone, which was used for flagging outliers and debiasing
the Downham Market record, we used additional historical total column ozone data for
several analyses. Specifically, we used total column ozone from various locations in Europe
(Brönnimann et al., 2003b) as well as a historical series from Canberra, (1929-1932), which
were digitised from daily values in Brönnimann et al. (2003a) and converted to the Bass-Paur
scale. While the European data, which were assumed to be of higher quality than some of the
other series, are available from the WOUDC, the other series described in Brönnimann et al.
(2003a) were only made available via an ftp site, which no longer exists. We therefore publish





all historical series used in this paper, together with all other series described in Brönnimann
et al. (2003a), in an electronic supplement to this paper (Table S1).
We also use a series from Aspendale near Melbourne, Australia, from the 1950s.
Observations with Dobson spectrophotometer #12 began in July 1955. Measurements were
taken near noon. Standard observational and calibration procedures were used (Funk and
Garham, 1962). The data since the IGY are today found in the WOUDC data base.
Concerning the earlier data, monthly means are found in various sources (Normand 1960,
Funk and Garham, 1962, as well as the scans from the Ozone Office), but the individual
values have so far not been published (the original data sheets are held at the National
Archives of Australia). We converted the data to the Bass-Paur scale using a scaling factor of

370    1.041.

For comparison with later periods (1990s and 2010s), we used total column ozone from the
WOUDC data base, namely from Lauder, NZ as well as Melbourne (measurements were
performed in the city in the 1990s and at the airport in the 2010s). All locations of the sites are
shown on Figure 1.
Further, we also used zonally averaged total column ozone data sets in order to embed the
Wellington series from the 1950s into a long term and global context. For the 1950s we use
the HISTOZ assimilated ozone data set (Brönnimann et al., 2013), which is based on an off-
line assimilation of historical total column ozone series into an ensemble of chemistry climate
model simulations (note that the monthly Aspendale data from 1955 onward have been
assimilated). For the 1990s we use the Zonal Mean Ozone Binary Database of Profiles
(BDBP, Bodeker et al., 2013) and for the 2010s we use the MOD7 release of the SBUV
(Version 8.6) merged total and profile ozone data set (Frith et al., 2014).
Comparisons were also performed with radiosonde and other upper-level data. We used
radiosonde data from IGRA2 (Durre et al., 2018) originating back to TD54 (see Stickler et al.,
2010). We used data from Auckland (1949-1957) for comparison with the Wellington ozone
data (at 490 km distance) and from Invercargill airport (1950-2020) for comparison with
Lauder ozone data for the period (1987-2010). Radiosonde data from Norfolk Island (1943-
2020) were also used for analysing spatial patterns. For the Downham Market data, no nearby
radiosonde station was available. We compared the total column ozone data with geopotential
height and temperature at all levels from the surface to the lower stratosphere. All three
stations were used to check the flow field for individual days. The locations of the stations are
also shown in Fig. 1.



Total column ozone data provide an excellent opportunity to assess the quality of upper-air
data sets. Brönnimann and Compo (2012) use total column ozone from the 1950s and 1960s
to assess the quality of the Twentieth Century Reanalysis data set version 2 (Compo et al.,
2011). This data set does not assimilate any upper air information, so it is interesting to know
how good the data agree with total column ozone observations. Additional data sets became
available in later years, including ERA20C (Poli et al., 2016). Hersbach et al. (2017)
produced a reanalysis for the period 1939-1963 assimilating historical upper-air data, termed
ERA-PreSAT, and compared it with 20CRv2 and ERA20C with respect to their correlation
with historical total ozone data in the period 1939-1963. Best correspondence was found with
ERA-preSAT, but no historical ozone data over Australia or New Zealand were used.
In the meantime, further data sets have become available, including CERA-20C (Laloyaux et
al., 2018) and 20CRv3 (Slivinski et al., 2019). Here we compare both historical total column
ozone data series with the three reanalysis data sets ERA-PreSAT, 20CRv3, and CERA20C.
For the processing, as in Brönnimann and Compo (2012) and Hersbach et al. (2017), all data
were deseasonalised by subtracting the first two harmonics of the seasonal cycle, and then
Pearson correlations were calculated. For the case of Downham Market, which only covers
one year, we fitted only the first harmonic function.

**4. Results**
*4.1. Downham Market*
We start the results with the shorter series of Downham Market, which is simpler as it allows
fewer comparisons. We first analysed correlations. Table 1 lists the correlations between the
re-evaluated Downham Market data (without the flagged values) and other total column
ozone series before and after deseasonalising. Note that for the reanalyses 20CRv3 and
CERA-20C, we used the ensemble mean.
Correlations are generally high. Even with the series of Arosa (at almost 1000 km distance), a
correlation of 0.78 was found (not shown). For the nearby Oxford series as well as for ERA-
preSAT, correlations exceed 0.90 on the absolute values and 0.75 on the anomalies. The
corresponding scatter plot (Fig. 6) for these two cases shows a linear relation with no apparent
deviations for high or low values. The 20CRv3 reanalysis, which in contrast to ERA-PreSAT
does not assimilate upper-level variables, also shows very high correlations. Slightly lower
correlations are found for CERA-20C.





We also analysed ozone fields for individual days. For this we supplemented the Downham
Market ozone observations with other observations from Europe, as given in Brönnimann et
al., 2003b). Five days were selected with good data coverage and pronounced positive or
negative anomalies of observed total column ozone over Downham Market. For these days,
observed ozone is plotted together with ozone from ERA-PreSAT (Fig. 7). We find a good
agreement between Downham Market and neighbouring stations as well as with ERA-
PreSAT total column ozone fields in all cases. In fact, most of the stations show a good
agreement, in this sense confirming the value of historical total column ozone data.

*4.2. Wellington*
For Wellington, in addition to similar analyses as for Downham Market, we also analysed the
series in a longer term context. Furthermore, we also compare the series with radiosonde data
from the stations displayed in Fig. 1.
Results of the correlation between Auckland radiosonde data and total column ozone in
Wellington are given in Table 2. For comparability purposes, we performed the same analysis
for a more recent period (1987-2010), with Invercargill radiosonde data and total column
ozone measurements in Lauder. From all series, the first two harmonics of the seasonal cycle
were subtracted, then the anomalies were correlated. As expected for a midlatitude site, we
find negative correlations with geopotential height at all levels, but strongest near the
tropopause and decreasing towards the surface and towards the stratosphere. For
temperatures, correlations change sign at the tropopause, i.e., high total column ozone is
related to a low tropopause altitude and to a cold upper troposphere and a warm lower
stratosphere.
Correlations are lower for the historical period than for the recent period. Differences could be
explained by the shorter spatial distance between Lauder and Invercargill (180 km) than
between Wellington and Auckland (490 km) and also the shorter temporal distance (in the
historical period radiosondes were launched once per day, first at 11 UTC, later at 0 UTC,
whereas in the second period we have twice daily soundings of which we chose the closer),
but also due to a lower quality of both data sources (ozone measurements and radiosonde).
Nevertheless, with correlations approaching -0.5 at the tropopause-level, results show that
day-to-day variability in total column ozone is likely to be well captured.
Next we compared Wellington ozone with ozone from reanalysis data sets (Table 3). Absolute
values of the reprocessed Wellington observations are 5.5% (adjusted coefficients) or 8%





(Bass-Paur) higher than those from the reanalyses. This is not due to outliers or specific
periods, but seems to be a feature of the bulk data. Correlations are lower than for Downham
Market, as expected since in the area of New Zealand, the reanalyses are not well constrained.
Nevertheless, we find correlations of around 0.6 to 0.8 for absolute values and of 0.45 for
anomalies. Lowest correlations on the anomalies are again found for CERA-20C. There is no
clear difference between the observation modes, except that the "infilled" daily data from the
Ozone Office are slightly worse (pointing to the value of working with original material).
As for Downham Market, we analysed some specific cases for Wellington. Figure 8 shows a
day with particularly high total column ozone in the series of Wellington. High ozone values
at midlatitudes are mostly due to upper-level troughs. The reanalyses ERA-PreSAT and
20CRv3 both reproduce higher ozone values related to an upper trough (100 hPa geopotential
height is also indicated), but do not reproduce the absolute value. 20CRv3 shows stronger
gradients in both fields.

*4.3. The long-term view*
Finally, we also put the reanalysed series from Wellington in a long term context (Fig. 9). We
compared the decadally averaged seasonal cycle for the 1950s (both for the Bass-Paur
coefficients and the adjusted coefficients) with that from Lauder from the 1990s
(corresponding to the peak of ozone depletion) and the 2010s. At least ten days were required
to form a monthly average from which decadal averages were then taken. Also shown on the
same figure are data from Aspendale/Melbourne for the three periods, and to the plot of the
first period we also added the Canberra, 1929-1932 series. Note that Canberra and Melbourne
are further north than Wellington, Lauder is further south. To make ozone at the different
latitudes comparable, we added offsets that were calculated from MOD7 zonal averaged data
(differences between the corresponding latitudes).
For the same three periods we also show zonal average total column ozone as a function of
latitude and calendar month in the assimilated total ozone data set HISTOZ (Brönnimann et
al., 2013; note that this data set does not assimilate the Wellington data) for the 1950s,
together with corresponding data from Bodeker et al. (2013) for the 1990s and from the
MOD7 SBUV merged data set for the 2010s. Note that the latitude-calendar month plots are
based on three different data sets. However, HISTOZ is by construction consistent with
BDBP, and the difference between MOD7 and BDBP is marginal.




For the 1950s, the shape of the curves agrees well, but there are considerable differences in
the levels, reflecting the uncertainty in absolute values. The Wellington curve with adjusted
coefficients is the lowest the Canberra series is (on average) the highest. Comparing the
figures for the 1950s and the 1990s, we find a large decrease between the two time periods.
This decrease is much stronger than the uncertainty between the data sets. Both in the station
data as well as in the global data set the change from the pre-ozone depletion climatology to
the maximum decade of ozone depletion, the 1990s, is thus clearly visible. Ozone depletion is
not just visible over Antarctica in spring, but also year round at southern midlatitudes and in
the subtropics. From the 1990s to the 2010s, a slight increase is seen at most latitudes in
MOD7, but hardly near 40° S. Likewise, only a faint increase is seen in the Lauder
observations.

**5. Discussion**
The re-evaluated total column ozone series from Wellington is internally consistent, although
its absolute level remains difficult to assess in absence of calibration information. From the
comparisons in Fig. 3 and assuming that in any comparison both series contribute roughly
equally to the error of the difference, a standard deviation of 13 DU in the difference between
two series is equivalent to a random error (standard deviation) of 9 DU in each of the two
series. We can therefore assume that in the reprocessed Wellington series the random error (in
terms of a standard deviation) is better than 10 DU. The systematic error is of approximately
the same magnitude. The choice of the absorption coefficients leads to a difference of 8.8 DU,
however, other uncertainties add to this. Comparisons with reanalysis data but also HISTOZ
suggest that the Wellington data are too high, but comparisons with Aspendale and Canberra
data (which are of a still lower quality, though) suggest that the data are too low. Too high
values could be due to calibration errors, or due to a too small aerosol correction. However,
high values are also possible for dynamical reasons such as a negative phase of the Southern
Annular Mode (SAM). In fact, pressure reconstructions indicate a sequence of years with
negative SAM in the 1950s (Fogt et al., 2011, 2016). In any case, we recommend using the
Wellington data with the adjusted coefficients, which best uses all information present to the
authors, although important pieces of information are lacking.
The Downham Market data are surprisingly precise, with a much higher correlation with
independent data than that data from Wellington. Also the absolute level is arguably better
determined as this series is statistically adjusted while the Wellington data are completely
independent from any other series. However, despite the good statistical performance, the





Downham Market data is of a different quality merely based on the fact that we do not have
raw data.
Both the Downham Market, UK, and Wellington, NZ, data well depict day-to-day variability,
which is closely related to the flow near the tropopause (Steinbrecht et al., 1998). This is
evidenced by the high correlation with radiosonde data in the case of Wellington and points to
a good quality of the ozone data. Note that lower correlations between total ozone and upper-
level variables are expected in the southern midlatitudes than at northern midlatitudes (see
Brönnimann and Compo, 2012). However, as we have no calibration information and no
intercomparison data, the series may not have trend quality.
For Downham Market, a large correction was necessary, but correlation with Oxford ozone
observations likewise suggests a high quality with respect to short-term changes, which is
surprising given the almost illegible data sheet. However, both the Oxford series and the
Donwham Market series might have been affected by tropospheric aerosols. This was the
reason why Dobson did not consider the Oxford series as very valuable for science, and the
same might also be the case for Downham Market.
Once the reliability of day-to-day variations in the ozone data is established, they can be used
to assess historical reanalysis products. In Brönnimann and Compo (2012), anomaly
correlations between observed and 20CRv2 ozone in Christchurch (in the 1920s) was found to
be around 0.5 (a similar value as for Wellington); for Europe anomaly correlations exceeding
0.6 were found. Hersbach et al. (2017) found anomaly correlations of 0.6 to 0.8 for total
column ozone in ERA-PreSAT, which is similar to what we find for Downham Market. We
find even higher correlations in our case, which might be due to better data but more likely
also reflect improvements in the reanalysis products.
Note that the quality of the Wellington data has not been tested for use in trend studies, and
we recommend not to use the data for trend analysis given the reported problems with the
instrument. Together with other data sources, the series nevertheless provides a glimpse at
ozone variability in the pre-ozone depletion era, which can be compared to later periods. All
data sources together illustrate a decrease in total column ozone from the 1950s to the 1990s,
approximately the time of minimum ozone (Solomon, 1999; Staehelin et al., 2001). An
increase is found in some data sets and stations since then and interpreted as a sign of ozone
recovery (Solomon et al., 2016). In the case of the southern midlatitude, an increase from the
1990s to the 2010s is hardly detectable. Historical data such as those from Wellington are
valuable as they depict ozone at southern mid-latitudes prior to the onset of ozone depletion.





Taken together, the data indicate that recovery is still far from complete. Values have not
nearly returned to the 1950s state.

**6. Conclusions**
Historical total column ozone data are relevant not just for analyses of long term changes in
the ozone layer, but also as a diagnostic of day-to-day atmospheric dynamics near the
tropopause. In this paper we present historical series from Wellington, New Zealand, 1951-
1959 and Downham Market, UK, November 1950 to October 1951. The data are re-evaluated
and analysed with respect to their quality. The former series will be made available via the
World Ozone and Ultraviolet Data Centre. Both series are published in the electronic
supplement, together with other historical total column ozone series used in this paper and
described in Brönnimann et al. (2003a).
The analyses reveal a good depiction of day-to-day variability, a fact which can be used to
assess the quality of reanalysis products, since the data cover a region and time period with
only few upper-air data. We show comparisons with the three reanalyses ERA-PreSAT
(which assimilates upper-air data), 20CRv3 and CERA20C, all of which show high
correlations, particularly over Europe, but also over New Zealand. Eventually, historical total
column ozone data could also be assimilated into historical reanalysis products.
The Wellington data were combined with other data sources to assess long-term ozone
changes over New Zealand. The 1950s in this context represent the era prior to the onset of
ozone depletion. Together, the data suggest that the recovery of the ozone is underway, but is
still far from the state it had in the 1950s. It should be noted, however, that the historical
Wellington data arguably do not have trend quality.

**Acknowledgements:** The Ozone Commission data sheets were provided to us by Alkis Bais. We wish to thank
the students at University of Bern who digitised the measurements.

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





**Figures**

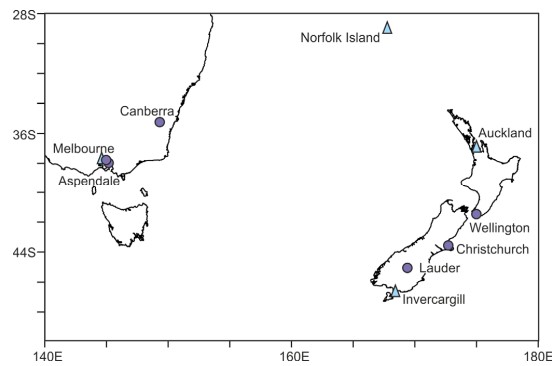



**Fig. 1.** Map of the stations used (circles: ozone, triangles: upper-air).





**Fig. 2.** Original data sheet from Wellington, NZ.




**Fig. 3.** New Years Card with data from Downham Market, 1950.



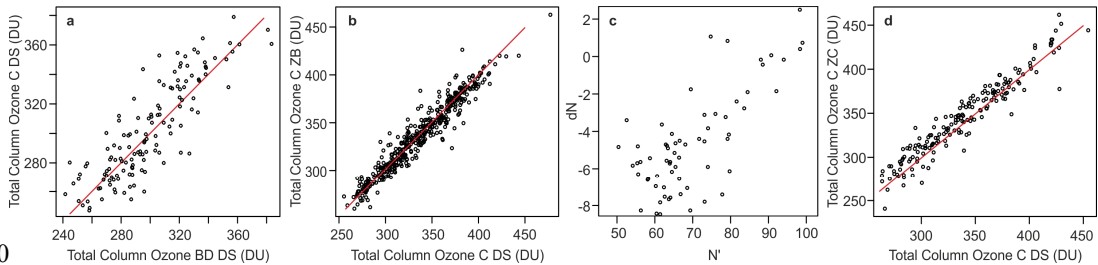


**Fig. 4.** Comparisons of (a) BD and C wavelength pair direct sun calculations, (b) fitted C ZB data against C DS
observations, (c) *dN* versus *N'* for C ZC observations and (d) reduced C ZC observations versus quasi-
simultaneous C DS observations. Here results are shown for the case with Bass-Paur absorption coefficients;
plots for the adjusted coefficients are indistinguishable. One-to-one lines are shown in red.

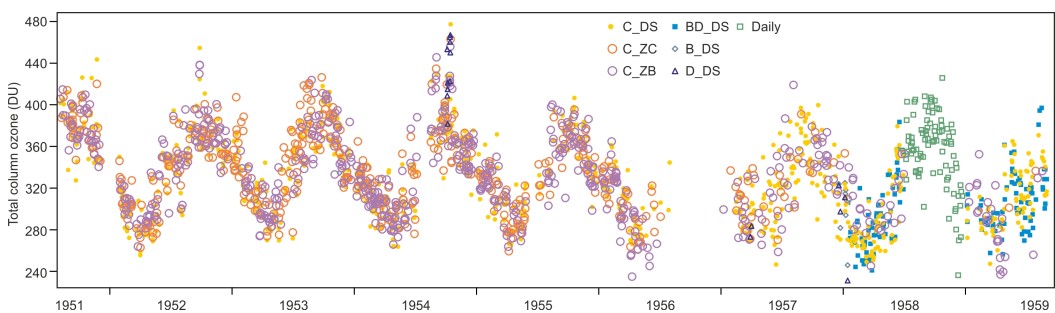

**Fig. 5.** Total column ozone at Wellington, 1951-1959 for different wavelength pairs and observation modes
(here for the case of Bass-Paur coefficients).

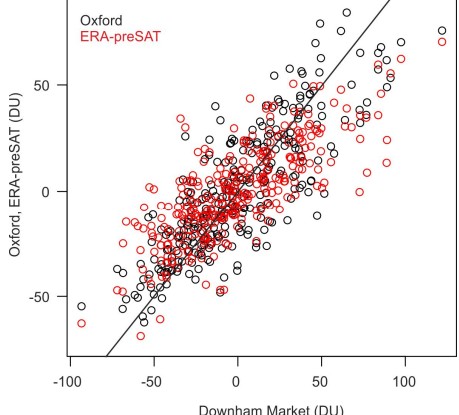

**Fig. 6.** Scatter plot of deseasonalised total column ozone data at Downham market against measurements
performed in Oxford as well as total column ozone data from the closest grid cell in ERA-PreSAT. The one-to-
one line is shown in black.

715

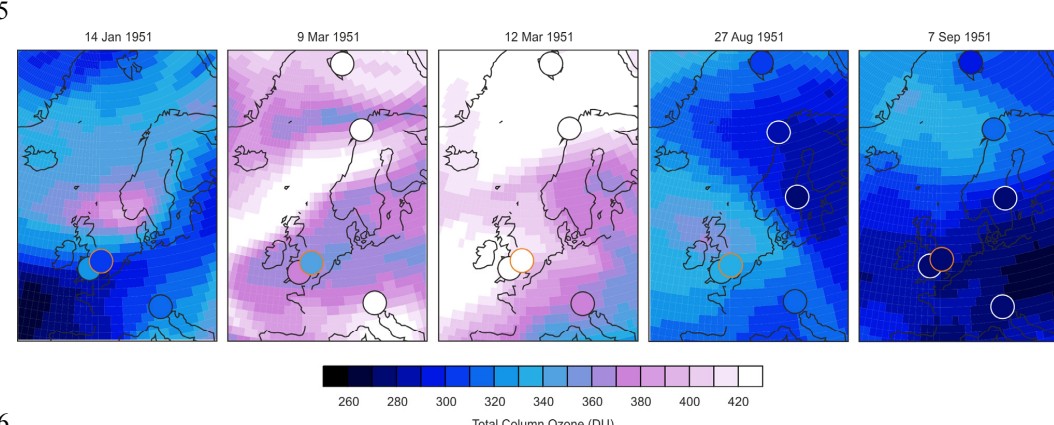

**Fig. 7.** Total column ozone in ERA-PreSAT as well as in observations from various stations on five days in the
year 1951 (Downham Market is marked with an orange outline of the circle).

719

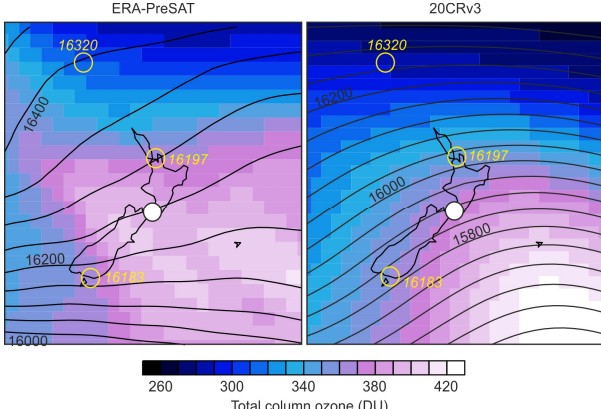

**Fig. 8.** Total column ozone and 100 hPa geopotential height on 25 Sep 1952 in ERA-PreSAT (left) and 20CRv3
(right). The filled circle indicates the measured total column ozone value at Wellington (434.6 DU, adjusted
coefficients), open circles indicate geopotential height from radiosonde (taken 12 hours later).

724

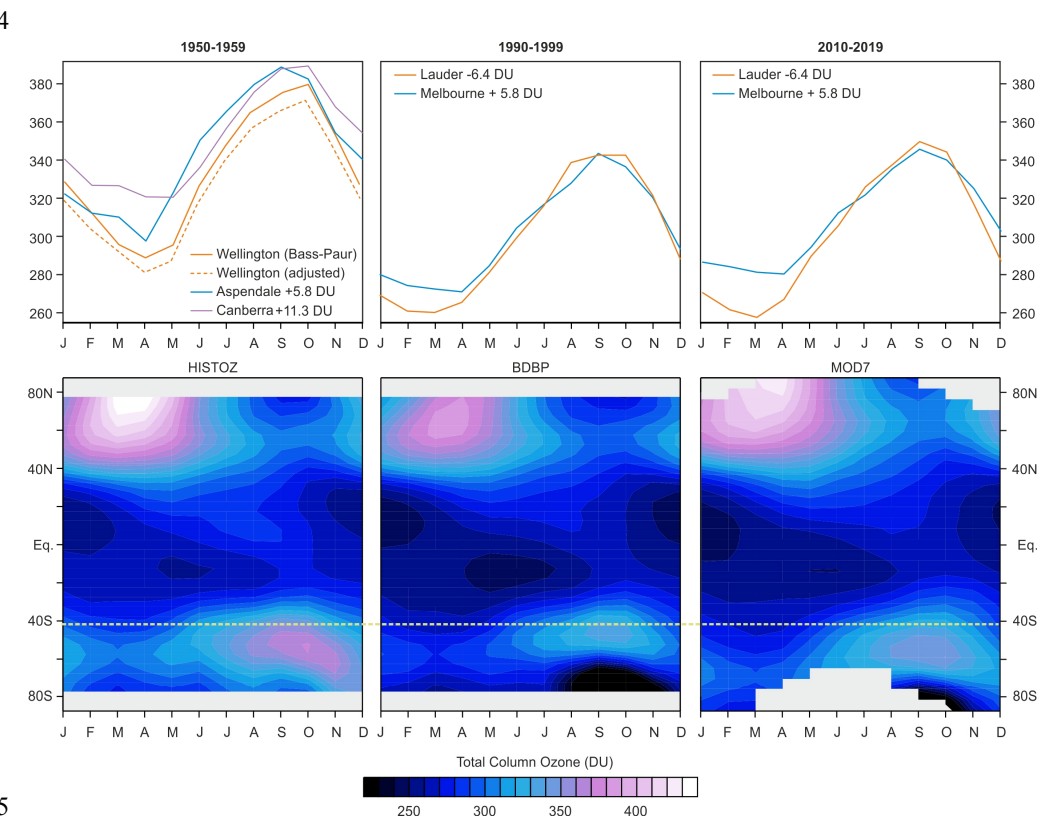

725

**Fig. 9.** Top: Decadally averaged annual cycle from total column ozone measurements in New Zealand and
Australia in the 1950s, 1990s, and 2010s. Note that the series are adjusted according to the annual mean offset
between the corresponding latitudes and that of Wellington in MOD7. Bottom: Zonally averaged total column
ozone as a function of calendar month and latitude in the data sets HISTOZ (1950s), BDBP (1990s) and MOD7
SBUV merge (2010s). The bottom left and middle panels are from Brönnimann (2015). Lauder and MOD7 data
end in 2018. The dashed line indicates the latitude of Wellington. Grey: No data.






**Tables**
**Table 1.** Pearson correlation coefficients of the re-evaluated total column ozone series from Downham Market
with other column ozone series. Anomalies refer to the values after subtracting the first harmonic function in
terms of day of year.

| Compared series | Absolute | Anomalies |
|---|---|---|
| Oxford | 0.91 | 0.83 |
| ERA-PreSAT | 0.90 | 0.75 |
| 20CRv3 ens. mean | 0.84 | 0.74 |
| CERA-20C ens. mean | 0.84 | 0.69 |


**Table 2.** Correlation coefficients (after deseasonalising) between total column ozone at Wellington and
radiosonde geopotential height and temperature at Auckland (1951-1957) as well as total column ozone at
Lauder and radiosonde data at Invercargill (1987-2010); see Fig. 1 for locations.

| p (hPa) | GPH | T | GPH | T |
|---|---|---|---|---|
| | Wellington | | Lauder | |
| 1000 | -0.22 | -0.18 | -0.17 | -0.44 |
| 850 | -0.28 | -0.35 | -0.34 | -0.50 |
| 700 | -0.35 | -0.40 | -0.43 | -0.56 |
| 500 | -0.42 | -0.41 | -0.53 | -0.59 |
| 400 | -0.44 | -0.40 | -0.56 | -0.58 |
| 300 | -0.46 | -0.25 | -0.59 | -0.51 |
| 200 | -0.45 | 0.16 | -0.60 | 0.28 |
| 100 | -0.33 | 0.42 | -0.40 | 0.69 |


**Table 3.** Correlation coefficients (before and after deseasonalising) between total column ozone at Wellington
and in other data sets (1951-1959) for different (the table relates to the case of Bass-Paur coefficient; results are
almost indistinguishable for the adjusted coefficients).

| | | all | C-DS | C-ZB | C-ZC | BD | Daily |
|---|---|---|---|---|---|---|---|
| ERA-PreSAT | abs | 0.65 | 0.66 | 0.65 | 0.68 | 0.71 | 0.66 |
| 20CRv3 | abs | 0.77 | 0.77 | 0.83 | 0.81 | 0.66 | 0.46 |
| CERA-20C | abs | 0.66 | 0.65 | 0.68 | 0.69 | 0.67 | 0.64 |
| ERA-PreSAT | anom | 0.44 | 0.45 | 0.45 | 0.48 | 0.51 | 0.36 |
| 20CRv3 | anom | 0.42 | 0.43 | 0.53 | 0.44 | 0.52 | 0.29 |
| CERA-20C | anom | 0.37 | 0.35 | 0.46 | 0.39 | 0.44 | 0.31 |
