# Peer review of "Total column ozone in New Zealand and in the UK in the 1950s"

_Atmospheric Chemistry and Physics, 2020_

## Referee Comment (RC1) · Anonymous Referee #3 · 7 Aug 2020

Brönnimann and Nichol describe two datasets of total column ozone records in the pre-IGY. One for Wellington, in New Zealand covering the period from 1951 to 1959 and the other from Downham Market in UK with about one year of data in 1951. The authors compare these datasets with data from other stations and models (ERA-preSAT, 20CRv3, CERA20c).

General comments:

These datasets belong to the pre-IGY period; thus they are a valuable source of information in O3 total column measurements history. The datasets itself are the main scientific value of this paper. Some considerations on the ozone before the 70's are also reported. The paper presents the results in a clear way, however I think that it can be slightly re-organized in order to help readability (I've added some suggestions be-

low). In addition, some results should be more clearly shown to fully support author's considerations on datasets quality (see some points in specific comments). The paper is written in a good English and the number of references is adequate. The supplement material is quite rich with Wellington, Downham Market and additional O3 datasets.

Specific comments:

1) Introduction: Can you add a description of other existing datasets (if any) like the one of Arosa?

2) Pag.2 line 61: remove D#17 is not relevant here ( it is just the code of the instrument) and replace it with Wellington coordinates. Do the same for Downham Market.

3) Pag.3 line 90: add C' wavelength values. This is also needed to understand why they have so low ozone influence (as stated in Pag.6 line 176).

4) Pag.5 line 161: Add wavelength for AD and BD, A . . .you can also add a table if you prefer. In general, I think you should provide more details on instrument configurations.

5) Pag.10: Please give more details on Ozone Office files and the ones from WOUDC (covered period, number of data, reference).

6) Pag.10 lines 311-313: "good agreement": please be more quantitative on the agreement, bias, the number of data used for this comparison or add a plot.

7) Pag.11 line 351: In my opinion the paragraph "Comparisons with . . ." should be moved into the results section. In addition, I find this section quite confusing, it is not really clear what you compare to what. Possibly it would be better to report the comparisons separately for Wellington and Downham Market in their respective subsections of section 4.

8) Pag.13 line 412: In my opinion the order should be maintained to help readability, Wellington before Downham Market.

9) Pag.13 lines 422-424: It would be nice to see these plots also.

10) Pag14 line 431-432: "good agreement": once again, please quantify.

11) Figure 4: Please add correlation and number of points on plots.

12) Figure5: Figure 5 is ok. However, I have a suggestion. Since the paper is on the two datasets (Wellington and Downham Market) and also the tile of the paper refers to both datasets, it would be better if you also show the Downham Market series, even if it is only one year of data. You may add a small panel on the left to this plot with the Downham Market time serie.

13) Figure 6: This plot is too qualitative. Please add correlations, bias, RMSE, number of points on plot. Possibly also the use of histograms and/or two different plots for the comparison with Oxford and ERA-presat instead of scattered plots should improve the quality of the plot and give a more quantitative idea of the agreement.

Technical comments:

1) Pag.4. line 111: add acronym for NIWA also here

2) Pag.4 line 118: Add coordinate

3) Pag.8 line 227: MICA: add acronym and reference

4) Table 1 and Table 3: this is just a suggestion, possibly you can replace "compared series" in Table 1 with "Downham Market vs". Something similar can be made in Table 3 by filling the first cell with "Wellington vs"

5) Data availability: As far as I understand from the abstract and conclusions, Wellington and Downham market datasets will be available from the World Ozone and Ultraviolet Data Centre (but they are also in the paper supplement). I suggest to add the direct link to WOUDC in the "Data availability" section in the final version.

---

## Referee Comment (RC2) · Bjoern-Martin Sinnhuber (Referee) · 9 Sep 2020

Stefan Brönnimann and Sylvia Nichol

**Bjoern-Martin Sinnhuber (Referee)**

bjoern-martin.sinnhuber@kit.edu

In this study, Brönnimann and Nichol re-evaluate and analyse Dobson total ozone observations at Wellington, New Zealand, and Downham Market, UK, for the pre IPY period. Great care is taken in a re-calibration of the data, as far as possible from the available sources. Historical ozone data are of great value, even if the data quality may be problematic for trend studies, as the day-to-day variability offers independent observations to test upper air re-analyses. The authors do this by correlating the total ozone data sets with results from different re-analysis projects.

The paper is well written and of interest to the readers of Atmos. Chem. Phys. The methods seem to be thorough and sound. Consequently I recommend publication in

[Figure]

Atmos. Chem. Phys. after consideration of the following, mostly minor, comments:

l.61: I suggest to spell out as "Dobson instrument #17"

l.151: For completeness it would be good to specify also the meaning of p and p0 in eq.1.

l.163: "Aerosol scattering can then be neglected" Suggestion: "...in which case eq.3 simplifies to:..." and then give the corresponding equation.

l.247: would be interesting how much this value differs from the standard value

l.287: "Z ZC" -> "C ZC"

l.328: "the all values" -> "all values"

l.381 and l.487-9: why not using SBUV MOD7 for the 1990s as well? Even if the differences are "marginal".

l.387: add distance Invercargill-Lauder for comparison here already (180km)

l.427: include "("

l.582: why not include names of students here?

l.744: something missing here: "for different"...?

Finally I would like to note that I support the idea of including the data, together with the other historic data sets, as a supplement to this paper.

---

## Author Comment (AC1) · 22 Sep 2020

**Reply to Reviewer 2**

l. 61: I suggest to spell out as "Dobson instrument #17"
Reviewer 1 suggested to omit this, so we move this from the title to the text and spell it out.

l. 151: For completeness it would be good to specify also the meaning of p and p0 in eq.1.
Thanks, this was an oversight (p is station pressure, p0 is sea-level pressure).
We will include this in the revised manuscript.

l. 163: "Aerosol scattering can then be neglected" Suggestion: "...in which case eq. 3 simplifies to:..."
and then give the corresponding equation.
Good suggestion – we will do this in the revised manuscript.

l. 247: would be interesting how much this value differs from the standard value
We will report the standard value in the new Table suggested by reviewer 1.

l. 287: "Z ZC" -> "C ZC"
Thanks.

l. 328: "the all values" -> "all values"
Thanks.

l. 381 and l. 487-9: why not using SBUV MOD7 for the 1990s as well? Even if the differences are
"marginal".
Since HISTOZ was generated with BDBP as standard, we prefer to plot this data set (also because it is
spatially more complete. Below is the plot for BDBP and MOD7. We will replace the formulation to
be more quantitative.

[Figure]

l. 387: add distance Invercargill-Lauder for comparison here already (180 km)
Thanks.

l. 427: include "("
Thanks.

l. 582: why not include names of students here?
Good suggestion – we will add the names to the Acknowledgements.

l. 744: something missing here: "for different"...?
...different wavelength and observation modes. Thanks, this will be changed.

---

## Author Comment (AC2) · 22 Sep 2020

**Reply to Reviewer 1**

1) Introduction:  Can you add a description of other existing datasets (if any) like the one of Arosa?
There is a long series from Tromsøe and there is Dobson's own record from Oxford going back to 1924 (although ending in the 1970s). A sentence on these two series will be added and corresponding references will be given.

2) Pag. 2 line 61: remove D#17 is not relevant here (it is just the code of the instrument) and replace it with Wellington coordinates. Do the same for Downham Market.
We remove D#17. Coordinates for Wellington are added (though not to the title, but the text). The same is done for Downham Market.

3) Pag. 3 line 90:  add C' wavelength values.  This is also needed to understand why they have so low ozone influence (as stated in Pag. 6 line 176).4) Pag. 5 line 161: Add wavelength for AD and BD, A... you can also add a table if you prefer. In general, I think you should provide more details on instrument configurations.
Thank you for this comment. Adding a Table is a good suggestion and will be done in the revised paper.

Table 1. Wavelengths (nm) and absorption and scattering coefficients for different wavelenth pairs for standard settings (Komhyr et al. 1993, 2007) and for the instrument in Kelburn

| Pair | short | long | α- α' | β-β' |
|---|---|---|---|---|
| A | 305.5 | 325.1 | 1.806 | 0.114 |
| B | 308.8 | 329.1 | 1.192 | 0.111 |
| C | 311.45 | 332.4 | 0.833 | 0.109 |
| C' | 332.4 | 453.6 | 0.040 | - |
| C (D#17) | 311.2 | 332.4 | 0.851 | 0.111 |
| D | 317.6 | 339.8 | 0.367 | 0.104 |

5) Pag. 10: Please give more details on Ozone Office files and the ones from WOUDC (covered period, number of data, reference).
More information will be given in the revied manuscript. Do you mean the PDF file from Environment Canada and the Archive Folder from the UK Met Office (cited in the paper as „Normand, 1961")? Both go back to the International Ozone Office and both are rather loose collections of data. The file from Environment Canada is a PDF-File with 1527 pages entitled „Early Total Ozone Information" and a data range on the title page given as 1959-1964 (which is incorrect as there are also earlier data). The „Normand, 1961" files were sent to me (SB) as photocopies of an archive folder by Stephen Farmer from the UK Met Office back in 2000. There is a large overlap between the two sources, but also unique matieral in each of them.
This will be detailed in the revised paper.

6) Pag. 10 lines 311-313: "good agreement": please be more quantitative on the agreement, bias, the number of data used for this comparison or add a plot.
We added in the text the correlations between our reworked data and those from the Ozone Office and WOUDC (after correcting for the date shift, but before excluding two outliers).

7) Pag. 11 line 351: In my opinion the paragraph "Comparisons with..." should be moved into the results section. In addition, I find this section quite confusing, it is notreally clear what you compare to what. Possibly it would be better to report the comparisons separately for Wellington and Downham Market in their respective subsections of section 4.
The intention of this paragraph was not to introduce results, but simply to report the data sets (and methods) used. However, we agree that it is not well written (in particular, the last two paragraphs contain a discussion of previous results, which should come later). In the revised paper we will change

the title of the section to „*Data sets used for comparisons*" and will shorten the last two paragraphs to only a list of data sets compared.

We will swap the results in the revised paper.

We will add a second scatterplot showing results of observations against 20CRv3 and CERA-20C (the figure is given below in our reply to comment 13).

We will change the sentences in the following way: „We find a good agreement between Downham Market and neighbouring stations as well as with ERA-PreSAT total column ozone fields in all cases (over the entire record, the standard deviation of differences is 25.9 DU). In fact, most of the stations show a good agreement (in the range of 30 DU), in this sense confirming the value of historical total column ozone data."

We will add numbers to the plots (all numbers are already given in the text).
4a: n = 136, r = 0.85, 4b: n = 429, r = 0.97, 4c: n = 65, r = 0.76, 4d: n = 178, r = 0.96

Thanks for this suggestion. We will show a figure that includes the Dowhnham Market data (see below).

[Figure]

We will add corresponding plots ffor all reanalyses and including the number of points in the plot, correlations, and RMSE (see below). We prefer scatterplots in order to better spot outliers or systematic behaviour. Comparison is also easier since the x-axis is the same (observations).

[Figure]

Technical comments:

1) Pag.4. line 111: add acronym for NIWA also here
Will do.

2) Pag. 4 line 118: Add coordinate
Will do.

3) Pag. 8 line 227: MICA: add acronym and reference
Will be added.

4) Table 1 and Table 3: this is just a suggestion, possibly you can replace "compared series" in Table 1 with "Downham Market vs". Something similar can be made in Table3 by filling the first cell with "Wellington vs"
Very good suggestion. This will be introduced in the revised manuscript.

5) Data availability: As far as I understand from the abstract and conclusions, Wellington and Downham market datasets will be available from the World Ozone and Ultraviolet Data Centre (but they are also in the paper supplement). I suggest to add thedirect link to WOUDC in the "Data availability" section in the final version.
We will try to do that in the final version if the link is available by then.

---

## Author Response (AR1)

**Reply to Reviewer 1**

1) Introduction: Can you add a description of other existing datasets (if any) like the one of Arosa? There is a long series from Tromsøe and there is Dobson's own record from Oxford going back to 1924 (although ending in the 1970s).
The corresponding sentence (l. 40-42) is now extended and reads: „Only few of the longer records were re-evaluated, such as those from Arosa (Staehelin et al., 1998), Tromsø (Hansen and Svenøe, 2005) and Oxford (Vogler et al. 2007)."

2) Pag. 2 line 61: remove D#17 is not relevant here (it is just the code of the instrument) and replace it with Wellington coordinates. Do the same for Downham Market.
We remove D#17. Coordinates for Wellington are added (though not to the title, but the text). The same is done for Downham Market.

3) Pag. 3 line 90: add C' wavelength values. This is also needed to understand why they have so low ozone influence (as stated in Pag. 6 line 176).4) Pag. 5 line 161: Add wavelength for AD and BD, A... you can also add a table if you prefer. In general, I think you should provide more details on instrument configurations.
Thank you for this comment. Adding a Table is a good suggestion and will be done in the revised paper.

Table 1. Wavelengths (nm) and absorption and scattering coefficients for different wavelenth pairs for standard settings (Komhyr et al. 1993, 2007) and for the instrument in Kelburn

| Pair | short | long | α- α' | β-β' |
|---|---|---|---|---|
| A | 305.5 | 325.1 | 1.806 | 0.114 |
| B | 308.8 | 329.1 | 1.192 | 0.111 |
| C | 311.45 | 332.4 | 0.833 | 0.109 |
| C' | 332.4 | 453.6 | 0.040 | - |
| C (D#17) | 311.2 | 332.4 | 0.851 | 0.111 |
| D | 317.6 | 339.8 | 0.367 | 0.104 |

5) Pag. 10: Please give more details on Ozone Office files and the ones from WOUDC (covered period, number of data, reference).
More information will be given in the revied manuscript. Do you mean the PDF file from Environment Canada and the Archive Folder from the UK Met Office (cited in the paper as „Normand, 1961")? Both go back to the International Ozone Office and both are rather loose collections of data. The file from Environment Canada is a PDF-File with 1527 pages entitled „Early Total Ozone Information" and a data range on the title page given as 1959-1964 (which is incorrect as there are also earlier data). The „Normand, 1961" files were sent to me (SB) as photocopies of an archive folder by Stephen Farmer from the UK Met Office back in 2000. There is a large overlap between the two sources, but also unique matieral in each of them.
In the revised paper we add the following sentences:
(l. 109-112) "...sent to the first author as a PDF file with 1527 pages (Bais, personal communication). The title of the folder is „Early Total Ozone Information" and a data range on the title page is given as 1959-1964; it nevertheless contains a number of earlier series, among them the Wellington and Downham Market data."
(l 135-138) "Photocopies of this archive folder were sent to the first author by Stephen Farmer (UK Met Office) in 2000. There is a large overlap between this file and the PDF File from Environment Canada, but there are also unique data in each of the folders."

6) Pag. 10 lines 311-313: "good agreement": please be more quantitative on the agreement, bias, the number of data used for this comparison or add a plot.

We added in the text the correlations between our reworked data and those from the Ozone Office and WOUDC (after correcting for the date shift, but before excluding two outliers).
(l. 322-323): "Correlations with the Ozone Office and WOUDC data amounted to 0.99 and 0.92, respectively."

7) Pag. 11 line 351: In my opinion the paragraph "Comparisons with..." should be moved into the results section. In addition, I find this section quite confusing, it is notreally clear what you compare to what. Possibly it would be better to report the comparisons separately for Wellington and Downham Market in their respective subsections of section 4.
The intention of this paragraph was not to introduce results, but simply to report the data sets (and methods) used. However, we agree that it is not well written (in particular, the last two paragraphs contain a discussion of previous results, which should come later).
In the revised paper we changed the title of the section to *„Data sets used for comparisons"* and shortened the last two paragraphs to only a list of data sets compared.

8) Pag. 13 line 412: In my opinion the order should be maintained to help readability, Wellington before Downham Market.
We swapped the results in the revised paper.

9) Pag. 13 lines 422-424: It would be nice to see these plots also.
We added a second scatterplot showing results of observations against 20CRv3 and CERA-20C (the figure is given below in our reply to comment 13).

10) Pag 14 line 431-432: "good agreement": once again, please quantify.
We changed the sentences in the following way:
(l 497-501) „We find a good agreement between Downham Market and neighbouring stations as well as with ERA-PreSAT total column ozone fields in all cases (over the entire record, the standard deviation of differences is 25.9 DU). In fact, most of the stations show a good agreement (in the range of 30 DU), in this sense confirming the value of historical total column ozone data."

11) Figure 4: Please add correlation and number of points on plots.
We added numbers to the plots (all numbers are already given in the text).

This is the new figure:

[Figure]

12) Figure5: Figure 5 is ok. However, I have a suggestion. Since the paper is on the two datasets (Wellington and Downham Market) and also the title of the paper refers to both datasets, it would be better if you also show the Downham Market series, even if it is only one year of data. You may add a small panel on the left to this plot with the Downham Market time serie.
Thanks for this suggestion. We will show a figure that includes the Dowhnham Market data (see below).

[Figure]

13) Figure 6: This plot is too qualitative. Please add correlations, bias, RMSE, number of points on plot. Possibly also the use of histograms and/or two different plots for the comparison with Oxford and ERA-presat instead of scattered plots should improve the quality of the plot and give a more quantitative idea of the agreement.

We added corresponding plots for all reanalyses and including the number of points in the plot, correlations, and RMSE (see below). We prefer scatterplots in order to better spot outliers or systematic behaviour. Comparison is also easier since the x-axis is the same (observations).

Below is the new plot:

[Figure]

Technical comments:

1) Pag.4. line 111: add acronym for NIWA also here
Done.

2) Pag. 4 line 118: Add coordinate
Done.

3) Pag. 8 line 227: MICA: add acronym and reference
Added.
(l. 236-237) „…the MICA (Multiyear Interactive Computer Almanac) software of the U.S. Naval Observatory"

4) Table 1 and Table 3: this is just a suggestion, possibly you can replace "compared series" in Table 1 with "Downham Market vs". Something similar can be made in Table3 by filling the first cell with "Wellington vs"
Very good suggestion.

This was changed in the revised manuscript.

5) Data availability: As far as I understand from the abstract and conclusions, Wellington and Downham market datasets will be available from the World Ozone and Ultraviolet Data Centre (but they are also in the paper supplement). I suggest to add thedirect link to WOUDC in the "Data availability" section in the final version.
We will try to do that in the final version if the link is available by then.

**Reply to Reviewer 2**

l. 61: I suggest to spell out as "Dobson instrument #17"
Reviewer 1 suggested to omit this, so we move this from the title to the text and spell it out.

l. 151: For completeness it would be good to specify also the meaning of p and p0 in eq.1.
Thanks, this was an oversight (p is station pressure, p0 is sea-level pressure).
We included this in the revised manuscript.
(l 161-162) „...$p$ and $p_0$ are station and mean-sea level pressure"

l. 163: "Aerosol scattering can then be neglected" Suggestion: "...in which case eq. 3 simplifies to:..." and then give the corresponding equation.
Good suggestion – we did this in the revised manuscript.

(l 170-171) Aerosol scattering can then be neglected and the equation reduces to:

$$X_{12} = \frac{(N_1 - N_2) - [(\beta - \beta')_1 - (\beta - \beta')_2]\dfrac{mp}{p_0}}{[(\alpha - \alpha')_1 - (\alpha - \alpha')_2]\mu} \qquad \text{(Eq. 4)}$$

l. 247: would be interesting how much this value differs from the standard value
We will report the standard value in the new Table suggested by reviewer 1.
(see reply to comment 3 of Rev. 1)

l. 287: "Z ZC" -> "C ZC"
Done.

l. 328: "the all values" -> "all values"
Done.

l. 381 and l. 487-9: why not using SBUV MOD7 for the 1990s as well? Even if the differences are "marginal".
Since HISTOZ was generated with BDBP as standard, we prefer to plot this data set (also because it is spatially more complete. Below is the plot for BDBP and MOD7. We replaced the formulation („marginal" is perhaps too strong, we now use „small") and quantify the differences in terms of standard deviations.
(l 467-469) "…the difference between MOD7 and BDBP is small. From 55° S to 60° N the standard deviation of the differences in zonally averaged, monthly total column ozone between the data sets is below 10 DU; the mean difference at 42.5° S amounts to 5.5 DU."

[Figure]

l. 387: add distance Invercargill-Lauder for comparison here already (180 km)
Thanks.

l. 427: include "("
Thanks.

l. 582: why not include names of students here?
Good suggestion – we will add the names to the Acknowledgements.
"We wish to thank Samuel Ehret, Michaela Mühl, Jerome Kopp, Juhyeong Han, Malve Heinz, Anita Fuchs, and Denise Rimer who digitised the measurements and Yuri Brugnara who organised the digitisation."

l. 744: something missing here: "for different"...?
Thanks, this was changed.
„...different wavelength and observation modes. „

[revised manuscript text omitted]

---

## Author Response (AR2)

**Reply to Reviewer 2**

Thanks for spotting the comments (all of which were due to the fact that we changed the sequence in the results part). All of the changes have been performed as indicated.